# Win Fast or Lose Slow: Balancing Speed and Accuracy in Latency-Sensitive Decisions of LLMs

**Hao Kang**
Georgia Institute of Technology
hkang342@gatech.edu

**Qingru Zhang**
Georgia Institute of Technology
qzhang441@gatech.edu

**Han Cai**
NVIDIA Corporation
hcai.hm@gmail.com

**Weiyuan Xu**
University of California, Berkeley

**Tushar Krishna**
Georgia Institute of Technology
tushar@ece.gatech.edu

**Yilun Du**
Harvard University
yilundu@gmail.com

**Tsachy Weissman**
Stanford University
tsachy@stanford.edu

## Abstract

Large language models (LLMs) have shown remarkable performance across diverse reasoning and generation tasks, and are increasingly deployed as agents in dynamic environments such as code generation and recommendation systems. However, many real-world applications, such as high-frequency trading and real-time competitive gaming, require decisions under strict latency constraints, where faster responses directly translate into higher rewards. Despite the importance of this latency–quality trade-off, it remains underexplored in the context of LLM-based agents. In this work, we present the first systematic study of this trade-off in real-time decision-making tasks. To support our investigation, we introduce two new benchmarks: **HFTBench**, a high-frequency trading simulation, and **StreetFighter**, a competitive gaming platform. Our analysis reveals that optimal latency–quality balance varies by task, and that sacrificing quality for lower latency can significantly enhance downstream performance. To address this, we propose **FPX**, an adaptive framework that dynamically selects model size and quantization level based on real-time demands. Our method achieves the best performance on both benchmarks, improving win rate by up to **80%** in Street Fighter and boosting daily yield by up to **26.52%** in trading, underscoring the need for latency-aware evaluation and deployment strategies for LLM-based agents. These results demonstrate the critical importance of latency-aware evaluation and deployment strategies for real-world LLM-based agents. Our benchmarks are available at Latency Sensitive Benchmarks.

## 1 Introduction

Large language models (LLMs) exhibit remarkable performance across various natural language processing (NLP) tasks and artificial intelligence (AI) applications, ranging from text generation to complex reasoning [OpenAI, 2023, Abdin et al., 2024, Team et al., 2025]. Beyond their standalone use, LLMs can be integrated into agent frameworks, enabling more sophisticated behaviors such as decision-making, multi-step reasoning, and planning [Yao et al., 2023, Shinn et al., 2023, Li et al., 2023, Du et al., 2023]. In these settings, a LLM acts as a decision-making agent, generating its actions or responses and then receiving feedback or rewards from environment. Many of these

39th Conference on Neural Information Processing Systems (NeurIPS 2025).

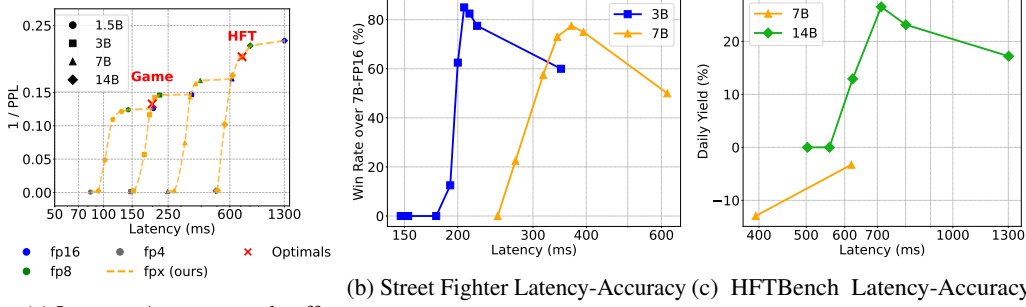

(a) Latency-Accuracy trade off

(b) Street Fighter Latency-Accuracy pareto curve

(c) HFTBench Latency-Accuracy pareto curve

Figure 1: Latency–accuracy trade-offs across different model configurations and tasks. (a) FPX enables a smooth and continuous trade-off between latency and accuracy, allowing models to meet diverse task-specific requirements. (b) In the Street Fighter benchmark, win rate first increases as latency decreases, peaking at a Pareto-optimal point, before dropping due to excessive accuracy loss. (c) Observation in HFTBench: daily yield improves with moderate latency reduction, but degrades when model accuracy is overly compromised.

agent tasks exhibit a high tolerance for inference latency, where slow responses are acceptable as long as the output quality remains high. Examples include code generation [Zhuo et al., 2024], mathematical problem solving [Xiao et al., 2023], and product recommendation [Wang et al., 2023], where correctness and completeness are prioritized over speed.

However, there is a different large class of real-time tasks that are highly sensitive to response latency and remains largely unexplored. These tasks often take place in dynamic environments that evolve continuously over time and are influenced by the agent's actions. In such settings, response latency becomes a critical factor in an agent's overall performance. Fast and well-timed actions are essential for obtaining positive rewards, while delays often leads to missed opportunities or suboptimal outcomes. One prominent example is gaming. Competitive games, such as Street Fighter [Su, 2010] and StarCraft [Samvelyan et al., 2019], take place in real-time environments where agents must perform multiple actions in a timely manner to win. Faster agents are more likely to stay synchronized with environmental changes and maintain an advantage, while slower agents may fall behind while processing outdated observations. Similarly, robotic control in dynamic physical environments demands rapid perception–action loops. Delayed responses in such settings, especially in high-stakes applications like autonomous driving, can result in unsafe or incorrect behavior.

Another important example is using LLM agents for high-frequency financial trading [He and Lin, 2022], where both low latency and high response quality are crucial. Stock exchanges match transactions based on real-time order flow, and faster trading agents can exploit arbitrage opportunities by acting before competitors respond. Prior research [Baron et al., 2019, He and Lin, 2022] in finance shows that trading latency directly impacts earning yields, motivating investment institutions to heavily invest in low-latency methods.

Across all these examples, both inference latency and response quality are critical for LLM agents to achieve strong performance. Either delayed or low-quality actions can lead to performance degradation or outright task failure. However, a fundamental trade-off exists between latency and quality when choosing models of different sizes or compressing them to low precisions. As illustrated by Figure 1a, larger models typically generate higher-quality outputs but suffer from longer inference time, while smaller or highly compressed models offer faster inference speed at the expense of reduced output quality. Therefore, as shown in Figures 1b and 1c, there exists an optimal solution for this trade-off and effectively balancing this trade-off is essential to optimize model performance in this type of real-world tasks.

In this paper, we are the first to systematically formulate and investigate the latency–quality trade-off in the context of real-time decision-making by LLM agents. We define this class of tasks as *latency-sensitive agent decision tasks*, where both output quality and response latency jointly determine the agent's overall performance. To evaluate model performance in such latency-sensitive setting, we develop two novel real-time evaluation benchmarks: (i) HFTBench: a high-frequency trading system tailored to assess real-time trading decisions of LLMs; (ii) StreetFighter: a competitive gaming platform that evaluate real-time gaming decisions of LLMs.

Based on our benchmarks, we observe that different tasks exhibit varying sensitivities to inference latency and output quality. As shown in Figure 1b and 1c, StreetFighter is more latency-sensitive and less quality-sensitive – timely, even if suboptimal, actions often lead to winning outcomes due to the game's simple yet rapidly evolving dynamics. In contrast, trading tasks demand both high quality and low latency. Inaccurate decisions can result in significant financial losses, making response accuracy as crucial as speed. Confronting such diverse task requirements, it is inherently challenging to identify the optimal point along the latency–quality trade-off. Existing approaches, such as selecting among fixed model sizes or applying static low-precision quantization, typically offer only a limited set of discrete options, which fail to capture the fine-grained trade-offs required across real-world tasks. To enable fine-grained searching, we introduce FPX, an adaptive mixed-precision inference framework that enables flexible control over inference latency while minimizing quality degradation. FPX jointly adjusts model size and dynamically mixes inference bitwidths across model layers to meet any specified latency target. Specifically, it hybridizes FP8 and FP4 inference kernels by selectively applying lower precision (FP4) to compression-tolerant layers while preserving FP8 for more sensitive components. This progressive and targeted quantization approach allows FPX to achieve continuous, fine-grained control across the latency–quality trade-off, effectively minimizing performance loss while satisfying diverse latency requirements. Our contributions are as follows:

- **Latency–Quality Trade-off.** We are the first to systematically formulate and investigate the latency–quality trade-off in the context of *latency-sensitive agent decision tasks*.
- **Latency-Sensitive Evaluation Benchmarks:** We introduce two novel benchmarks for evaluating LLM performance in the latency-sensitive settings: (1) a high-frequency trading (HFT) system specifically tailored to LLMs, and (2) a competitive fighting game environment based on *Street Fighter* from the DIAMBRA platform [Palmas, 2022].
- **Adaptive Mixed Precision Inference Framework.** We propose an adaptive mixed-precision inference framework that that enables flexible control over inference latency while minimizing quality degradation.

## 2 Background

### 2.1 Low Precision Inference to Reduce Latency

Recent advancements in hardware-supported low-precision inference, such as FP8 and FP4 [Micikevicius et al., 2022, Li et al., 2025], offer significant improvements in both throughput and latency over standard full-precision inference (FP16). These methods employ floating point quantization (FP Quant) to map high-precision tensors to low-precision ones, reducing memory footprint of both model weights and activations [Li et al., 2025]. Given a tensor $X$, FP quantization rescales its entries and rounds them to values within a bounded range determined by bitwidth $b$:

$$Q(X) = \text{round}\left(\frac{X}{\text{scale}_X}\right), \quad \text{scale}_X = \begin{cases} \frac{\max(|X|)}{\text{range}_b} & \text{if } \max(|X|) > \text{range}_b \\ 1 & \text{otherwise} \end{cases} \quad (1)$$

Here, $Q(X)$ is the quantized matrix and $\text{range}_b$ is determined by the bitwidth $b$, specifically 240 for FP8 [Micikevicius et al., 2022] and 6 for FP4. During inference, the forward pass in linear layers can be approximated as:

$$XW \approx \text{scale}_X \cdot \text{scale}_W \cdot Q(X)Q(W) \quad (2)$$

As supported by hardware, low-precision inference benefits from faster floating-point operations, improved memory bandwidth, and efficient datatype conversion. With substantially reduced memory footprint, low-precision inference can significantly lower end-to-end inference latency compared to FP16. For instance, FP8 typically provides up to $2\times$ latency speedup while maintaining near-lossless output quality, making it widely adopted. FP4, on the other hand, can yield up to $4\times$ latency reduction, but often causes severe degradation in model performance, limiting its standalone application. Recent work such as SVDQuant [Li et al., 2025] attempts to mitigate the accuracy loss by combining it with low-rank corrections and smoothing. However, such approaches remain static and do not offer adaptive control over the latency–quality trade-off in real-time, latency-sensitive tasks.

### 2.2 Additional related work on throughput optimization

Another line of related work focuses on conventional serving scenarios, whose primary goal is to improve serving throughput while maintaining near-lossless performance. For example, systems such

as vLLM [Kwon et al., 2023] and SGLang [Zheng et al., 2024] achieve around $6.4\times$ throughput improvements without compromising output quality. While such systems may reduce latency in specific conditions (e.g., shared prefill structures in SGLang), they are generally not designed to optimize latency in a task-specific manner. Other efforts, such as AI Metropolis [Xie et al., 2024], build distributed cluster systems to accelerate agentic simulations through speculative execution of multiple agents. These approaches aim to maximize simulation throughput but are not tailored for latency-sensitive, real-world agent deployments. Separately, a substantial body of work explores integer quantization to improve serving throughput [Lee et al., 2024, Frantar et al., 2023, Kang et al., 2024b, Zirui Liu et al., 2023]. Unlike hardware-supported FP quantization, integer quantization typically requires highly costly dequantization operations during inference. While it enables larger batch sizes and improves overall throughput, the dequantization overhead significantly limits its effectiveness in reducing latency [Lin et al., 2024, Zhao et al., 2024, Kang et al., 2024a]. Other works [Tang et al., 2023, Pandey et al., 2023] propose mixed-precision schemes combining integer and floating-point formats to balance throughput and accuracy. However, these methods remain static and lack the ability to provide fine-grained, dynamic control over LLM inference latency.

# 3 Latency-Sensitive Agent Decision Tasks

In this section, we formally define the *latency-sensitive agent decision tasks*, formulate its *latency-quality trade-off*, and introduce two real-time evaluation benchmarks: (i) *HFTBench* – a high-frequency trading system tailored to evaluate real-time trading decisions of LLMs, and (ii) *StreetFighter* – a competitive gaming environment that assess real-time gaming decision of LLMs.

## 3.1 Formulating Latency-Sensitive Agent Decision Tasks

Consider a general setup of in which an LLM agent interacts with an environment $\mathcal{E}$ to solve a task. At time step $t$, the agent receives an environmental observation $o_t \in \mathcal{O}$. After spending $\Delta_t$ time conducting inference, the agent responds an action $a_{t+\Delta_t} \in \mathcal{A}$ following its decision policy $\pi_\theta$:

$$a_{t+\Delta_t} \sim \pi_\theta(c_t) \quad \text{where} \quad c_t = \{(o_0, a_{0+\Delta_0}), (o_1, a_{1+\Delta_1}), \ldots, o_t\}. \tag{3}$$

Here $c_t$ is the context to the agent, for example, a conversation between a user and the agent.

As introduced in Section 1, conventional agent tasks exhibit high tolerance to LLM inference latency $\Delta_t$. A simple case is single-step tasks such as one-hop question answering [Kwiatkowski et al., 2019] or document summarization [Shaham et al., 2022], where the agent generates a single action given an initial input prompt $o_0$, and the outcome is evaluated purely based on the output quality: $r = \mathcal{R}(a|o_0)$, where $\mathcal{R}$ denotes a task-specific evaluation or reward function. A more complex case involves multi-step task-solving, such as multi-step mathematical reasoning or code generation, where the agent produces a sequence of actions over time. In such tasks, the overall performance depends on the cumulative quality of all outputs:

$$r = \sum_t \mathcal{R}(a_{t+\Delta_t}|c_t) \tag{4}$$

In both cases above, the environment is relatively static, and delayed responses are acceptable as long as the agent maintains high response quality. Correctness is prioritized over speed.

However, many real-world tasks, such as gaming, robotic control, and high-frequency trading, take place in dynamic environments $\mathcal{E}_t$ that evolve rapidly over time. This setting remains large unexplored and we name it as *latency-sensitive agent decision tasks*. In these tasks, a delayed actions $a_{t+\Delta_t}$ is often rendered ineffective or obsolete by the time it is executed under the updated environment state $\mathcal{E}_{t+\Delta_t}$, leading to missed opportunities or degraded outcomes. In such setting, the agent is evaluated not only by *what* it decides, but also by *how long* it decides. To succeed, it must produce actions that are both high-quality and timely. The reward thus becomes a function of both the decision and its latency, evaluated under the evolved environment:

$$r = \sum_t \mathcal{R}(a_{t+\Delta_t}|\mathcal{E}_{t+\Delta_t}). \tag{5}$$

This formulation captures the core challenge of latency-sensitive tasks: enabling LLM agents to make fast and accurate decisions in environments where speed is as critical as accuracy.

## 3.2 HFTBench: High-Frequency Trading Benchmark

**Latency and Quality in Financial Trading.** High-frequency trading (HFT) involves rapidly submitting buy and sell orders to centralized exchanges, where transactions are strictly matched based on arrival time. In this setting, even millisecond-level differences in reaction latency can significantly impact profitability. Temporary arbitrage opportunities often arise when short-term imbalances cause the bid–ask spread to widen. Agents that respond quickly can capitalize on these brief windows by buying at temporarily depressed prices or selling at elevated ones—before the market rebalances.

However, latency alone is insufficient. High-quality trading decisions rely on correctly interpreting market conditions, which often require processing multi-step patterns in historical prices, order book dynamics, and occasionally external signals such as policy announcements or financial news. While smaller LLMs benefit from lower latency, our experiments show that they often fail to capture such complex financial patterns, resulting in poor decisions that negate their speed advantage.

**Benchmark Design.** We construct a realistic backtesting simulation using historical per-second trading data from Polygon.io [Polygon.io, 2024]. Each agent receives synchronized market observations at 1-second intervals and must decide whether to take action. To isolate the effect of decision latency, all agents have access to the same information and observation windows.

When an arbitrage opportunity is detected, agents initiate inference. The simulated exchange ranks agents by their response time and assigns execution prices accordingly: faster agents secure more favorable prices. We implement a linearly decaying price model of time and price, where trading advantage diminishes with slower responses—mimicking real-world queue-based order execution.

**Evaluation Protocol.** Each agent observes a compact state containing prior execution prices, current bid–ask margins, available capital, and time remaining in the trading session. To avoid unnecessary LLM calls, inference is only triggered when the bid–ask margin exceeds a preset threshold $b$. Agents are evaluated by their cumulative daily yield, and a configurable cooling window $t$ is applied between evaluations to improve simulation efficiency.

## 3.3 Gaming Benchmark: Street Fighter

**Latency Sensitivity in Competitive Games.** In real-time competitive games such as *Street Fighter* and *StarCraft*, delayed actions can result in immediate penalties, positional disadvantages, or even round losses. Unlike financial trading, where both decision quality and latency play important roles, these games are overwhelmingly latency-sensitive. In our experiments(Figure 1b), agents with just a 20% reduction in response time consistently outperform their slower counterparts. Interestingly, the strategic depth of *Street Fighter* is relatively limited, and well-prompted small LLMs can produce effective actions, provided they respond quickly enough.

**Benchmark Design.** We build on top of DIAMBRA's simulation platform [Palmas, 2022] to support real-time *Street Fighter* matches with local model inference. To improve performance for compact models (e.g., <7B parameters), we augment the prompt with tailored few-shot examples specific to each character and scenario. This enhancement helps mitigate performance degradation from reduced model capacity.

**Evaluation Protocol.** Agents receive a concise game state that includes character-specific move sets, recent action history, and a contextual prompt. We evaluate performance using the ELO rating system [Elo, 1967], where agents compete across multiple matches against a diverse set of opponents. ELO scores are updated dynamically to reflect win–loss outcomes, providing a stable and interpretable metric for real-time decision quality under latency constraints.

## 3.4 Discussion

**LLM Agents in Finance and Gaming.** Recent works have explored the application of LLM-based agents in both financial trading and competitive gaming. In finance, FinMem [Yu et al., 2023] and FinAgents [Zhang et al., 2024] demonstrate that LLM agents outperform traditional reinforcement learning and rule-based strategies. This performance gain is attributed to the robustness of LLMs against overfitting and their unique ability to process unstructured inputs, such as policy updates or financial news, through in-context learning. However, these approaches are evaluated on static historical datasets and ignore the role of response timing, which is crucial in real-time trading. In contrast, our high-frequency trading benchmark captures not only the agent's decision quality, but

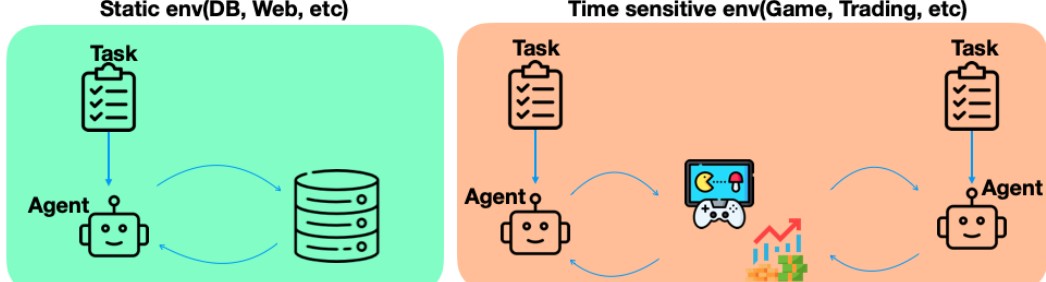

Figure 2: Comparison of agentic LLM for Static environments like code generateion or research and time sensitive environments like trading and gaming. Environment is constantly changing with time and other agent's interaction. For such tasks, reward is related to both quality and latency of agents.

also its response speed and the pricing gap it can exploit —offering a more faithful simulation of real-world trading dynamics.

In the gaming domain, prior work has applied LLM agents to real-time strategy and fighting games such as *StarCraft* and *Street Fighter* [Ma et al., 2024, Palmas, 2022]. These studies primarily focus on improving action quality and designing robust inference pipelines. However, they do not consider the inherent trade-off between latency and decision quality that governs real-time decision performance. Our benchmarks specifically emphasize this trade-off, providing a clearer understanding of how timing impacts success in latency-sensitive environments.

## 4 FPX: Adaptive Mixed Precision Inference Framework

In this section, we introduce FPX, our adaptive mixed-precision inference algorithm designed for *latency-sensitive agent decision tasks*. As motivated in section 1, FPX dynamically adjusts precision at the operator level, switching between the matrix multiplication kernels of FP8 and FP4, to enable continuous fine-grained control over the latency–quality trade-off.

### 4.1 Adaptive Mixed-Precision Algorithm Design

The core goal of FPX is to balance latency and accuracy by selectively lowering the precision of only the most compression-tolerant components in a model. Instead of modifying full models or entire layers, we adopt a more granular precision control scheme that applies FP4 only to linear layers that can tolerate aggressive quantization, while preserving FP8 for more sensitive parts.

To ensure compatibility with a wide range of transformer architectures, we focus exclusively on optimizing matrix multiplication operators, which dominate inference latency in LLMs. These include query/key/value (QKV) projections, output projections, and feedforward layers. Other components, such as normalization and attention mechanics, are left untouched to maintain functional correctness and deployment simplicity.

Importantly, because transformer linear layers exhibit similar structural and computational properties, latency gain from replacing FP8 with FP4 is approximately uniform across layers. This decouples precision assignment from latency impact and shifts the optimization focus entirely toward minimizing quality loss. To quantify the robustness of each linear layer to quantization, we compute a relative error metric $\varepsilon_l$ based on activation outputs under FP16 and FP4 execution:

$$\varepsilon_l = \frac{\|A_l^{\mathrm{fp16}} - A_l^{\mathrm{fp4}}\|_2}{\|A_l^{\mathrm{fp16}}\|_2} \tag{6}$$

Here, $A_l^{\mathrm{fp16}}$ is the output of layer $l$ under FP16 execution, and $A_l^{\mathrm{fp4}}$ is the output when the same input is processed using an FP4 kernel. The normalized error $\varepsilon_l$ captures the fidelity loss introduced by low-precision inference and serves as the basis for selecting compression candidates.

Given a user-specified compression ratio $\gamma \in [0, 1]$, we define a precision assignment function $\delta(l) \in \{4, 8\}$ for each linear layer $l$:

$$\delta(l) = \begin{cases} 4 & \text{if } l \in \mathcal{S}_\gamma \\ 8 & \text{otherwise} \end{cases}, \quad \text{where } \mathcal{S}_\gamma = \operatorname*{argmin}_{\substack{S \subset \mathcal{L} \\ |S| = \gamma L}} \sum_{l \in S} \varepsilon_l \tag{7}$$

---

**Algorithm 1** Adaptive FP4/FP8 Precision Assignment for Transformer Layers

---

**Require:** Transformer model $\mathcal{M}$ with $L$ linear layers $\mathcal{L} = \{l_1, \ldots, l_L\}$, calibration dataset $\mathcal{D}$, compression ratio $\gamma \in [0, 1]$
**Ensure:** Precision assignment function $\delta(l) \in \{4, 8\}$ for all $l \in \mathcal{L}$
 1: **for all** layer $l \in \mathcal{L}$ **do**
 2:     Run FP16 inference on $\mathcal{D}$ to collect outputs $A_l^{\text{fp16}}$
 3:     Simulate FP4 output $A_l^{\text{fp4}}$ using the same inputs
 4:     Compute relative quantization error:

$$\varepsilon_l = \frac{\|A_l^{\text{fp16}} - A_l^{\text{fp4}}\|_2}{\|A_l^{\text{fp16}}\|_2}$$

 5: **end for**
 6: Sort layers in ascending order of $\varepsilon_l$
 7: Select $\mathcal{S}_\gamma$ as the $\gamma L$ layers with the smallest $\varepsilon_l$
 8: **for all** layer $l \in \mathcal{L}$ **do**
 9:     **if** $l \in \mathcal{S}_\gamma$ **then**
10:         $\delta(l) \leftarrow 4$                    ▷ Assign FP4 to quantization-tolerant layers
11:     **else**
12:         $\delta(l) \leftarrow 8$                    ▷ Preserve FP8 for sensitive layers
13:     **end if**
14: **end for**
15: **return** $\delta$

---

Here, $\mathcal{S}_\gamma$ denotes the subset of $\gamma L$ layers with the lowest quantization error. This design ensures that FP4 is selectively applied to the most robust layers, enabling substantial latency gains while minimizing quality degradation.

### 4.2 Offline Calibration

To compute the layer-wise quantization error $\varepsilon_l$, we perform a one-time offline calibration using a held-out language modeling dataset. Following standard practice in quantization research [Xiao et al., 2024, Hooper et al., 2024], this calibration phase estimates typical activation distributions observed during inference. Concretely, we run full-precision (FP16) inference on the Wikitext-2 dataset [Merity et al., 2016], capturing both the input and output activations for each linear layer. Then, we simulate FP4 execution by running each layer individually, replacing its FP16 kernel with an FP4 kernel, while keeping all other layers unchanged. This isolates the quantization impact at the layer level and yields a reliable estimate of $\varepsilon_l$ for each candidate.

The complete precision assignment pipeline is summarized in Algorithm 1.

## 5 Experiments

We evaluate our method on two time-sensitive task benchmarks introduced in Section 3. We then perform an ablation study to analyze the lantecy-quality trade-off brought by FPX.

### 5.1 Experimental Setup

**Models.** To ensure a fair comparison and reduce the complexity of the search space, we conduct our experiments on a family of models pretrained on similar datasets. Evaluating across heterogeneous model families could introduce biases due to differences in pretraining quality, architecture, or tokenizer design. Therefore, we focus on the Qwen2.5 model suite [Qwen et al., 2025], ranging from 1.5B to 14B parameters.

**Benchmark Configurations.** For the high-frequency trading (HFT) benchmark, we evaluate on stock data from Nvidia and Amazon on August 5th, 2024. We follow the configuration introduced in Section 3.2, setting the profit threshold $b$ to 2% and the time window $t$ to 1 minute. The initial cash for agent is 10,000 dollars. For the gaming benchmark, we conduct 40 matches between model pairs and compute win rates to derive ElO ratings.

**Method Configurations.** We discretize the compression ratio $\gamma$ of FPX into steps of 0.1 to explore the trade-off between latency and accuracy across different benchmarks. We only report the **best-performing setting** for each model in each task. In our experiments, fine-grained changes in $\gamma$

Table 1: Evaluation results on latency-sensitive benchmarks. Our method achieves the best latency–reward trade-off across both tasks. Only shows top-6 results. More results are shown in appendix.

| HFTBench | | | |
|---|---|---|---|
| **Model Parameter Size** | **Bitwidth Avg** | **Latency (ms)↓** | **Daily Yield (%)↑** |
| 14B (ours) | 7.2 | 713 | **26.52** |
| 14B | 8 | 801 | 23.14 |
| 14B | 16 | 1302 | 17.20 |
| 7B | 16 | 619 | -3.28 |
| 7B (ours) | 7.6 | 386 | -7.25 |
| 7B | 8 | 394 | -12.94 |

| Street Fighter | | | |
|---|---|---|---|
| **Model Parameter Size** | **Bitwidth Avg** | **Latency (ms)↓** | **ELO Score↑** |
| 3B (ours) | 6.8 | 195 | **5.99** |
| 7B (ours) | 7.2 | 354 | 2.33 |
| 3B | 8 | 222 | 2.19 |
| 3B | 16 | 349 | 0.25 |
| 7B | 8 | 394 | -0.44 |
| 1.5B | 8 | 142 | -1.25 |

generally have minimal effect, indicating that our selected settings are near-optimal. All experiments are run on an RTX 5090 GPU unless otherwise specified. 14B models are served across multiple GPUs using model parallelism.

## 5.2 Baseline Techniques

Time-sensitive benchmarks are sensitive to both model quality and inference latency. Any quantization method that results in slower inference than FP8 is excluded from consideration. We evaluate the following baselines:

• *FP16*: A standard dense model with both activations and weights in 16-bit floating point. This serves as the upper baseline in quality but incurs the highest latency.

• *FP8*: A widely adopted low-precision format for Hopper and newer GPU architectures, representing both activations and weights as 8-bit floating point. It typically offers near-lossless accuracy with significantly better efficiency than FP16.

• *FP4*: A highly compressed representation where both activations and weights are quantized to 4 bits and packed as 8-bit integers. This setting drastically improves efficiency but at the huge cost of model response quality. It is only available on blackwell architecture GPUs.

## 5.3 Evaluation Result

Table 1 demonstrates that FPX, by dynamically trading off latency and quality through adaptive model size and bitwidth selection, achieves the highest daily yield on HFTBench and the best overall reward across both benchmarks.

**High-Frequency Trading (HFTBench).** This benchmark requires a careful balance between latency and response quality. We observe that larger models, such as 14B, outperform smaller alternatives due to their stronger ability to recognize profitable opportunities. In contrast, smaller models often fail to detect high-reward patterns or generate outputs that are too unreliable to be translated into effective trading decisions. FPX improves the latency of the 14B model by compressing 20% of its linear layers into FP4, while preserving FP8 for the rest. This enables a favorable speed–quality trade-off, allowing 14B+FPX to achieve the highest daily yield among all candidates. Interestingly, we find that further reducing the latency of weaker models like 7B actually harms performance. Faster response does not help if the decisions themselves are poor, and can even increase the rate of loss.

Table 2: Performance under different compression levels on Qwen2.5 models for HFTBench and Street Fighter."−" means model performance is complete destroyed.

| HFTBench – Qwen2.5-14B | | | |
|---|---|---|---|
| Gamma ($\gamma$) | Latency (ms)↓ | PPL↓ | Daily Yield (%)↑ |
| 0.0 (FP8) | 801 | 4.55 | 23.14 |
| 0.2 | 713 | 4.92 | **26.52** |
| 0.4 | 623 | 6.71 | 12.93 |
| 0.6 | 558 | – | 0.00 |
| 0.8 | 503 | – | 0.00 |
| 1.0 (FP4) | 489 | – | 0.00 |
| **Street Fighter – Qwen2.5-3B+FPX versus Qwen2.5-3B-FP16** | | | |
| Gamma ($\gamma$) | Latency (ms)↓ | PPL↓ | Winrate (%)↑ |
| 0.0 (FP8) | 222 | 6.85 | 72.5 |
| 0.2 | 207 | 7.03 | 77.5 |
| 0.3 | 200 | 9.02 | **80.0** |
| 0.4 | 192 | 11.59 | 62.5 |
| 0.6 | 178 | 17.42 | 12.5 |
| 0.8 | 153 | – | 0.0 |
| 1.0 (FP4) | 147 | – | 0.0 |

**StreetFighter.** This task is highly latency-sensitive, yet quality still matters. Our method achieves the best performance with a 3B model configured with 30% of layers in FP4 and 70% in FP8. Notably, although the fastest candidate, the 1.5B model with full FP8 inference, has the lowest latency, it performs poorly due to its limited decision-making capability. Moreover, the environment itself imposes an upper bound on effective response rate. In StreetFighter, each character action takes a fixed amount of in-game time to complete, with an effective frame rate of around 5 actions per second (i.e., 200ms per action). Any optimization that reduces model latency beyond this threshold yields no further benefit, as the game cannot process actions faster than this limit.

### 5.4 Ablation Study

**Latency–Quality Trade-off of FPX** We evaluate the Pareto frontier of the latency–quality trade-off induced by FPX across both benchmarks and bitwidth configurations. Specifically, we apply FPX to the Qwen2.5 model family and compare against standard FP16 inference. Our results show that FPX effectively adapts each model's inference path between FP8 and FP4 regimes, dynamically balancing latency and accuracy. Notably, the optimal trade-off point varies by task and model: for instance, on HFTBench with the 14B model, the best performance is achieved at $\gamma = 0.2$, while on Street Fighter with the 3B model, the optimal setting is $\gamma = 0.3$. These findings highlight that latency-sensitive decision-making tasks require task-specific latency–quality configurations, and FPX enables LLM agents to navigate this trade-off effectively.

## 6 Limitations and Conclusion

In this work, we present the first systematic study of the latency–quality trade-off for LLM-based agents in *latency-sensitive agent decision tasks*. To support this investigation, we introduce two real-time evaluation benchmarks: **HFTBench**, a high-frequency trading simulator, and **StreetFighter**, a competitive gaming environment. In both settings, rapid yet accurate decisions are essential to achieving high downstream rewards.

To meet the heterogeneous demands of these tasks, we propose FPX, an adaptive mixed-precision inference framework that dynamically adjusts model precision to optimize for task-specific latency–quality trade-offs. By selectively applying FP4 quantization to compression-tolerant layers while retaining FP8 for sensitive components, FPX enables fine-grained latency control with minimal performance degradation.

Extensive experiments on Qwen2.5 model variants demonstrate that FPX consistently discovers favorable operating points that outperform fixed-precision baselines across both domains. Our ablation results further reveal that the optimal compression configuration varies significantly by task and model, underscoring the importance of latency-aware deployment strategies for LLM agents.

While FPX demonstrates strong empirical gains, it has limitations. Our current precision assignment operates at the layer level for simplicity and compatibility. More fine-grained schemes, such as token-level precision control, may unlock better trade-offs, but require significantly more complex implementation and kernel support. We left this optimization for future works.

We hope that our benchmarks and findings encourage future research toward building efficient, adaptive LLM systems and algorithms that prioritize latency-awareness in real-world applications, rather than focusing solely on maximizing accuracy or model performance.

# 7 Acknowledgement

We are deeply grateful to Professor Tsachy Weissman for his guidance. We also thank Jinyan Su(PhD student from Cornell University) for valuable suggestions on refining the paper.

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

## A    Visualization of HFTBench Data

Here we provide the high-low price per second for the data we have used for HFTBench tests in Figure 3. Red rectangle points out the buy-sell price gap in short time, which provide trading opportunity for agents. Such opportunity only happens in short time. Buying and selling decisions of other agents will decrease the gap quickly in miliseconds.

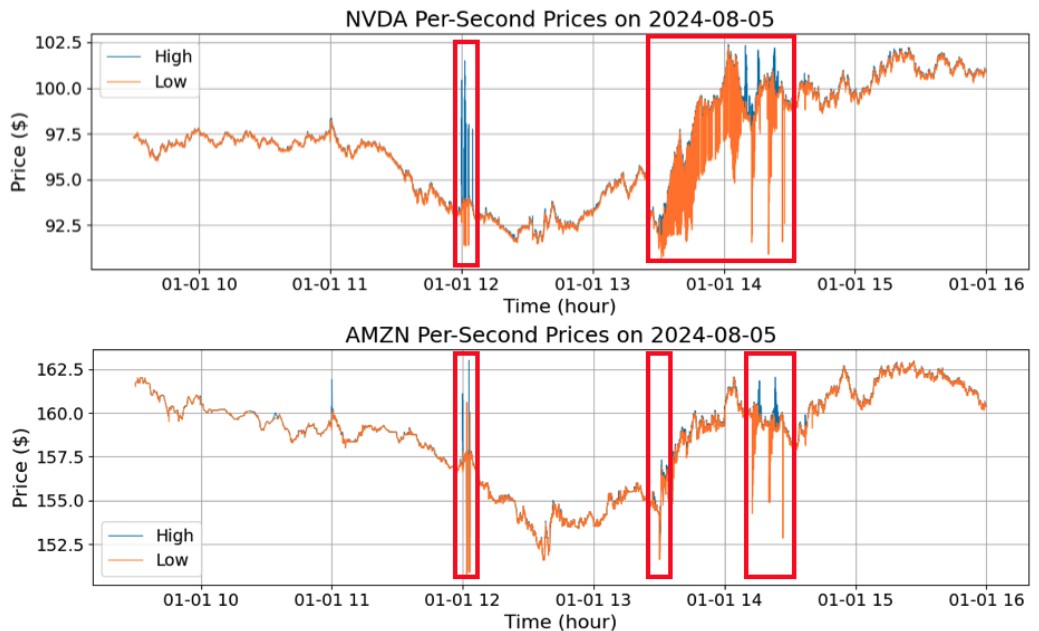

Figure 3: Visualizations of HFTBench testing data.

## B    More Experiment Results for StreetFighter

Here we provide more results of StreetFighter. Competitors are run for 40 round and calculate the ELO scores.

Table 3: Latency and yield comparison on **StreetFighter**.

| Model Parameter Size | Bitwidth Avg | ELO Score(%)↑ |
| --- | --- | --- |
| 3B | 6.8 | **5.65** |
| 3B | 7.2 | 3.57 |
| 7B | 6.8 | 2.33 |
| 7B | 7.2 | 2.33 |
| 3B | 8 | 2.18 |
| 3B | 16 | 0.26 |
| 7B | 8 | -0.45 |
| 1.5B | 16 | -1.25 |
| 1.5B | 8 | -2.66 |
| 7B | 16 | -2.89 |
| 14B | 8 | -3.14 |
| 14B | 16 | -5.94 |

## C    Latency Profiling of Quantization method

We conduct a detailed latency profiling of various quantization methods on RTX 5090 GPUs. For the 14B model, we employ model parallelism across two GPUs. The results are summarized in Table 4. Our findings show that both FP8 and FP4 kernels yield substantial latency reductions compared to

the FP16 baseline. However, for the W4A16 configuration, where model weights are stored as 4-bit integers, the latency benefits are less pronounced, except in large models such as Qwen2.5-14B. This is likely due to the overhead introduced by data type conversion and dequantization. These results suggest that hybrid usage of FP8 and FP4 kernels is a promising strategy for improving inference efficiency, particularly on large-scale models.

Table 4: Latency (ms) Comparison Across Quantization Schemes

| Model | FP16 | FP8 | W4A16(int) | FP4 |
|---|---|---|---|---|
| Qwen-1.5B | 203 | 142 | 254 | 83 |
| Qwen-3B | 349 | 222 | 323 | 147 |
| Qwen-7B | 619 | 394 | 537 | 248 |
| Qwen-14B (2×5090) | 1302 | 801 | 792 | 492 |

