# OpenReview forum: "Win Fast or Lose Slow: Balancing Speed and Accuracy in Latency-Sensitive Decisions of LLMs"
_NeurIPS.cc/2025/Conference — NeurIPS 2025 spotlight_

### Official Review · Reviewer_mdup · 2025-06-19

**Clarity:** 3
**Significance:** 3
**Originality:** 2
**Rating:** 4
**Confidence:** 3

**Summary:**

This paper investigates the trade-off between inference latency and output quality in LLM-based agents, especially under real-time constraints. The authors identify that many applications (e.g., high-frequency trading, real-time gaming) require rapid decision-making, where even minor latency can degrade performance. To address this, they introduce two latency-sensitive benchmarks: HFTBench (for high-frequency trading) and StreetFighter (for competitive gaming). To optimize performance across diverse latency requirements, the authors propose FPX, an adaptive mixed-precision inference framework. FPX adjusts both model size and bitwidth (mixing FP8 and FP4) dynamically, applying lower precision to more quantization-tolerant layers. Experiments show that FPX achieves improved win rate and financial yield compared to static baselines, with up to 80% higher win rate in StreetFighter and 26.52% yield improvement in HFTBench.

**Questions:**

1. What is the most practically beneficial factor in your framework that enables LLM agents to operate more effectively in a border range of latency-sensitive tasks?
2. How does FPX perform on real hardware that supports FP4 inference? Are there actual deployment results beyond simulation?
3. Can you discuss whether any new low-precision design choices were made that specifically benefit latency-critical scenarios beyond standard mixed-precision strategies?
4. Could you elaborate on how the precision configuration is applied during inference? Is it dynamically adjusted per input, per timestep, or fixed at deployment time? How feasible is this control pipeline in real-world systems with limited runtime flexibility?

**Ethical Concerns:**

["NO or VERY MINOR ethics concerns only"]

**Final Justification:**

Nice rebuttal. It resolves my concern. So I change my score to align with the other reviewers.

**Limitations:**

Yes, limitation is clearly stated in the conclusion section.

**Quality:**

3

**Strengths And Weaknesses:**

Strengths
+ Well-motivated by latency-sensitive applications like trading and gaming.
+ Introduces two task-specific, real-time benchmarks.
+ Empirically demonstrates that adjusting the latency–accuracy balance can improve downstream performance.

Weaknesses
- It is unclear whether the observed performance improvements translate to practical deployment gains in real-world latency-sensitive decision systems.
- FP4 is only simulated, and the impact of actual hardware behavior is not evaluated, especially across non-Blackwell architectures.
- The adaptive quantization method closely follows known practices (e.g., layer-wise calibration), with limited novelty in the core algorithmic contribution beyond application to LLM agents.

---

> ### Author Rebuttal · Authors · 2025-07-30
>
> We thank the reviewer for your valuable suggestions for our paper. We would like to confirm that **all of our experiments are conducted on NVIDIA RTX 5090 GPUs with FP4 hardware support but not simulated**.
> ## Question1: What is the practically beneficial factor of FPX?
> The most practically beneficial factor in our FPX framework is its ability to provide a **continuous and fine-grained trade-off between generation accuracy and latency (illustrated clearly in Figure 1(a)).** Unlike previous compression algorithms that only support **discrete or very limited accuracy–latency trade-off points**, FPX allows **adaptive, layer-wise precision selection (FP8 and FP4 combinations in our experiments) tailored explicitly for latency-sensitive tasks**.
>
> In latency-sensitive tasks, latency directly impacts the downstream reward and evaluation metrics(in section 3). **Different time sensitive tasks require different latency and accuracy to achieve the optimal point.** Hence, being able to apply a carefully tuned lossy optimization strategy that precisely meets task-specific demands is crucial. For example, gaming scenarios can benefit substantially from more aggressive precision reduction (lower latency at the cost of accuracy), while trading applications demand higher accuracy at moderately reduced latency. For example, in Table 2 in our paper, the smaller model with progressive quantization performs the best in StreetFights. While at the same time, bigger models with near-lossless perform well at HFTBench.
>
> FPX tackles this challenge by automatically calibrating and selecting near-optimal precision settings, **ensuring strong LLM performance across diverse latency-sensitive tasks**.
> ## Question2: How does FPX perform on real hardware?
> We understand the concern of the reviewer of our framework, but we would like to clarify that **none of the FP4 results in our paper are simulated**. All FP4 inference results, including latency and accuracy measurements, **were obtained from actual hardware experiments conducted on NVIDIA RTX 5090 GPUs with cuda kernel written by ourselves (as stated in Section 5.1 and Table 4 of our paper)**. NVIDIA's Blackwell architecture officially supports FP4 inference, as confirmed in the Blackwell documentation. Rubin, NVIDIA’s next-gen hardware, is confirmed to support FP4 inference as well at GTC2025. FP4 is going to be a common practical low-bit precision like FP8.
> ## Question3: What is the new low-precision design of our paper?
> We propose two key innovations beyond standard mixed-precision strategies, tailored for latency-critical applications:
> ### 1. Offline Calibration for **Continuous Latency-Accuracy Trade-offs** (Section 4.2, Algorithm 1):
>
>  We introduce a new offline calibration algorithm that estimates per-layer quantization error under FP4/FP8 execution by measuring the relative degradation in activation outputs. Based on this, we compute a precision assignment δ(l) that selectively applies FP4 to compression-tolerant linear layers and FP8 elsewhere.
>
> Prior calibrations (e.g., SmoothQuant) focus on **near-lossless optimization**, not on **adaptive latency–accuracy trade-offs required by real-world latency-aware agents**. In contrast, FPX supports **fine-grained, continuous control** over latency and accuracy, beyond fixed global settings in past approaches. This enables **task-specific adaptation**, identifying Pareto-optimal configurations (e.g., γ = 0.2 for HFT, γ = 0.3 for Street Fighter), as shown in Table 2 and Section 5.4.
>
> ### 2.Custom Mixed Precision Inference Kernel with Automatic Dtype Fused Output Transfer (Implementation detail in Section 4.1):
> Mixed-precision inference introduces **overhead** when **consecutive layers use different bitwidths**. For example, an FFN where the first layer runs in FP8 and the second in FP4. Prior implementations typically output FP16 and perform a separate conversion for the next layer, **incurring latency**, especially in single-batch inference.
>
> To address this, we implement a custom FP8/FP4 kernel that fuses output quantization with memory storage, directly converting SMEM results into the target dtype (FP8/FP4) before writing to HBM. This eliminates separate dtype conversion and memory copy passes, **improving inference speed by at least 10% in mixed-bitwidth layers on RTX 5090.** Such fusion is essential in latency-critical scenarios where small overheads accumulate across many layers. FP4 is a new feature recently supported by NVIDIA this year. To our best knowledge, we are the **first** to implement kernel fusion between FP8 and FP4.
> ### Table R5T1: Ablation study of kernel fusion on RTX5090.
> | Model |      Bitwidth Avg      | Fusion Speed up|
> | :---: | :--------: | :------: |
> | Qwen2.5-14B |       8       |    100%     |
> | Qwen2.5-14B |   7.6   |     112.7%     |
> | Qwen2.5-14B |   7.2   |     119.3%     |
> | Qwen2.5-14B |   6.8   |     123.4%     |
>
>
> ### 3. In addition, a core contribution of our work is the introduction of the first set of benchmarks specifically designed for latency-sensitive agent tasks, HFTBench and StreetFighter.
>
> These benchmarks allow us to systematically study the **fundamental trade-off between latency and accuracy**, which is often overlooked in prior LLM evaluations. By doing so, we highlight a previously **underexplored challenge**: that in real-world deployments (e.g., trading, gaming), the performance of LLM agents is **not solely determined by output quality, but also by their response time**. This insight poses **a new systems and algorithm-level problem**, which is how to dynamically balance accuracy and latency under strict timing constraints.
>
> ## Question4: FPX deployment and fitness in real-world system.
> As discussed in Algorithm 1 of the paper, FPX first performs **offline calibration** on wikitext dataset to rank each layer's sensitivity to FP4 compression and get the precision assignment function. It then **adjusts the compression ratio γ(percentage of FP4 layers) according to the function** to achieve the desired latency-accuracy trade-off by running and calibrating on real sensitive task benchmarks. The parameter γ, as well as the precision assignment function, **is fixed at inference time.**
>
> The deployment of FPX, combining FP8 and FP4 precision, is applicable to numbers of devices: it can run on any hardware supporting FP8 and FP4 inference. Currently, NVIDIA's Blackwell GPUs and certain AMD edge devices support FP4, and NVIDIA's upcoming Rubin GPUs will also provide FP4 compatibility. **This shows that FP4 will be a common low bit precision supported by hardware like FP8.**
>
> ## Weakness1: Practical deployment gain on real-world systems.
> We would like to clarify that all benchmarks in our benchmarks are grounded in **real-world data** designed to closely reflect practical deployment settings and our experiments are conducted on **real hardware**.
>
> Specifically, StreetFighter is implemented using the DIAMBRA simulation engine, which mirrors the behavior of real-time fighting games and is deployable on personal computers. HFTBench, as detailed in Section 3.2, is constructed using real historical per-second market data from Polygon.io (e.g., NVDA, AMZN on 2024-08-05) and simulates trading under realistic latency-sensitive order execution rules.
>
> This is a **backtesting-based evaluation**, not a synthetic simulation, where agents receive time-synchronized observations and are ranked by response latency, with execution prices decaying linearly as in **real-world** limit order books.
>
> While real-world latency-sensitive systems **vary in dynamics and constraints**, this motivates FPX to be **adaptive and general-purpose**. Rather than relying on fixed precision or handcrafted settings, FPX supports **continuous latency–accuracy trade-offs via tuning γ**. This flexibility allows engineers to adjust configurations for the latency budget of any target system, including trading, gaming, or robotics mentioned by other reviewers as well.
>
> In short, our study not only uses realistic benchmarks based on real data, but also proposes a solution, FPX, that is broadly applicable across heterogeneous latency-sensitive domains with performance improvements.
>
> ## Weakness2: FP4 is simulated.
> We would like to clarify that FP4 inference in our experiments is **not simulated**.
>
>  All results reported are obtained on RTX 5090 GPUs, which natively support FP4 inference. Our FP4 kernels are implemented and executed on the device, not emulated or simulated in software.
>
> To further demonstrate cross-hardware generalizability, we additionally evaluate FPX on NVIDIA GB200, which also supports native FP4 execution. As shown in Table R5T2, FPX configurations with bitwidth averages below 8 continue to outperform the FP8 baseline, achieving up to 75% win rate on StreetFighter, validating that our method is effective **across multiple FP4-enabled architectures**.
> ### Table R5T2: StreetFighter Ablation study of GB200.
> | Model |      Bitwidth Avg      | Win rate vs. FP16|
> | :------: | :-------------: | :--------: |
> | Qwen2.5-32B(ours) |       7.2       |    75%     |
> | Qwen2.5-32B(ours) |   7.6   |     70%     |
> | Qwen2.5-32B |   8   |     62.5%     |
>
>
> ## Weakness3: Method lacks of novelty.
> We respectfully disagree with the claim that our method lacks algorithmic novelty. While FPX builds on layer-wise calibration, its key innovation is in **adapting this to real-time agent settings**, where latency directly affects the reward (see Equation 5).
>
> We propose a **continuous latency–accuracy trade-off mechanism** via a tunable parameter γ, enabling **inference-time latency control without retraining**, which is a feature **not supported** by prior quantization methods focused on static LLM serving.
>
> We are also extending FPX to domains like **autonomous driving**, where early results show its effectiveness in managing sensor-to-action delay, highlighting its value as a general-purpose adaptive inference framework.

---

> > ### Comment · Reviewer_mdup · 2025-08-01
> > **Thank you for the detailed response**
> >
> > Thank you for the rebuttal. It resolves my concern, so I raised my score accordingly.

---

### Official Review · Reviewer_oRvQ · 2025-06-30

**Clarity:** 3
**Significance:** 3
**Originality:** 3
**Rating:** 4
**Confidence:** 3

**Summary:**

This work investigates real-time sequential decision making by LLM-based agents,
a scenario where both latency and quality of actions matter.
It first proposes two benchmarks, one for high-frequency trading and the other for the StreetFighter game.
Then, it proposes FPX, a quantization method that allows choosing different quantization levels for adapting the same LLM to the needs of different tasks.
Empirical evaluations are conducted for such time-sensitive environments, where models of different sizes and quantization levels are compared.

**Questions:**

I'd suggest adding some concrete examples of prompts and responses in the proposed benchmarks. Is the LLM prompted to answer with the action directly (with a response length of just one or two tokens) for each action, in order to achieve latency within hundreds of ms?

**Ethical Concerns:**

["NO or VERY MINOR ethics concerns only"]

**Final Justification:**

The authors have addressed my concerns, and I would maintain my rating.

**Limitations:**

Yes, the authors have discussed limitations of this work.

**Quality:**

2

**Strengths And Weaknesses:**

**Strengths:**


* This work studies a nice topic (i.e., latency-sensitive decision making by LLM agents) and proposes interesting benchmarks, pointing to promising directions for further research.

* This work provides insights for LLM-based agent in latency-sensitive scenarios. One example is Eq (5), which points out that the reward depends on the rapidly changing environment. Another insight is that smaller latency for each action implies faster reaction to environmental change, and the ability to execute more actions within a fixed period of time, which can be crucial for latency-sensitive scenarios.

* The proposed FPX mixed-precision quantization method allows the same model to adapt to different tasks by selecting the appropriate quantization levels.




**Weaknesses:**



* One limitation of FPX, as acknowledged by the authors in Sections 5.4 and 6, is that its hyperparameter gamma need to be tuned carefully for each <task, model> tuple  in order to achieve satisfactory performance.

* The authors answer "yes" to Question 7 of the checklist regarding statistical significance. This is incorrect, as each number in the experiments is the result of a single trial (unless I missed something).



* Writing can be further polished, for example:
    * Typo: Line 261, "lantecy" --> "latency"
    * Line 85 in the introduction where the authors claim the main contributions of this work: to be more precise, it should be "LLM-based agent" rather than just "agent".



**Disclaimer:**

Due to my limited familiarity with certain literature, I'm not sure about
* whether this work is indeed "**the first** to systematically formulate and investigate the latency–quality trade-off in the context of real-time decision-making by LLM agents" (Line 60);
* how novel and original is Algorithm 1 (which seems fairly natural and straightforward);
* how reasonable and realistic is the benchmark design / problem formulation for quant trading (Lines 173 - 180).

I believe that the AC and other reviewers can make better judgments on this aspects than I do.

---

> ### Author Rebuttal · Authors · 2025-07-30
>
> ## Question1: Concrete example of prompts
> Thanks for your suggestion! Sure, we force LLM to answer with action directly for now to decrease the decoding step for better latency. The max generated token is set to 32 tokens.
> Here we provide an example prompt of our StreetFight Benchmark.
> ```
> You are the best and most aggressive Street Fighter III 3rd strike player in the world.
> Your character is {character}. Your goal is to beat the other opponent.
> if you are far from opponent, use Move Closer and Fireball more often.
>         If you are close to opponent or already move closer, try to use Punch and Kick more often.Megapunch, Hurricane, and other combinations uses more time but are more powerful. Use them when you are close to opponent and you are getting positive scores or winning. If you are getting negative scores or losing, try to Move away and use Kick.
> The moves you can use are:
> {move_list}
> ----
> Example 1:
> Context:
> You are very far from the opponent. Move closer to the opponent. Your opponent is on the left.
> Your last action was Medium Punch. The opponent's last action was Medium Punch.
> Your current score is 108.0. You are winning. Keep attacking the opponent.
>
> Your Response:
> - Move closer
> - Move closer
> - Low Kick
>
> Example 2:
> Context:
> You are close to the opponent. You should attack him.
> Your last action was High Punch. The opponent's last action was High Punch.
> Your current score is 37.0. You are winning. Keep attacking the opponent.
> Your Response:
> - High Punch
> - Low Punch
> - Hurricane
> Now you are provided the following context, give your response using the same format as in the example.
>
> ```
> One of the answers generated by Qwen2.5-3B is:
> ```
> - Low Punch
> - High Punch
> - Move Away
>
> ```
> Prompts are specially designed for HFTBench as well.
> ## Weakness1: The Hyperparameter gamma needs to be tuned for each model and task.
> We appreciate the reviewer’s observation and agree that the compression ratio γ is a task- and model-dependent hyperparameter that influences the latency–accuracy trade-off.
>
>  However, we would like to emphasize that our primary contributions lie in **(1) defining and constructing realistic latency-sensitive benchmarks, and (2) systematically evaluating the trade-offs across models, bitwidths, and γ values on these benchmarks (see Sections 3 and 5).**
>
> By exposing the Pareto frontier for various <task, model> configurations, our work provides the **first comprehensive empirical understanding of how inference latency interacts with decision quality in real-time agentic settings**. This understanding is crucial and was **previously missing from the literature**.
>
> While automated γ tuning or runtime adaptation is an important direction, we explicitly position this as **future work** (Section 6). Our goal in this paper is to **establish the problem setting, introduce an interpretable, practical framework (FPX), and demonstrate its effectiveness across representative real-world scenarios (trading and gaming)**. This lays the foundation for more sophisticated tuning or control mechanisms in follow-up work.
>
> ## Weakness2: Statistical evaluation.
> We thank the reviewer for pointing this out. You are correct. Our initial submission reported single-trial results without error bars, and we acknowledge this mistake in answering Checklist Question 7.
>
> To address this, we have now repeated experiments with multiple independent runs and report the mean ± standard deviation across these runs. We will incorporate these revised results (with appropriate error bars) into the final version of the paper. StreetFighter runs 10 times. Each time consists of 40 rounds of fighting for each model pair. And we run HFTBench 4 times indepdently.
> ### Table R4T1: StreetFighter ELO Scores with std
> | Model parameter size |      Bitwidth Avg      |   ELO(40 rounds)   |
> | :----------------------------: | :---------------------: | :---------------------: |
> | 3B(ours)                 |           6.8                |    5.92 ± 0.13            |
> | 7B（ours)               |            7.2              |       2.31 ±  0.17         |
> | 3B                           |            8               |         2.21  ±  0.05        |
> | 3B                           |            16               |       0.26  ±  0.09        |
> | 7B                           |            8               |     -0.44   ±   0.12         |
> | 1.5B                       |            8               |      -1.26   ±  0.07          |
>
> ### Table R4T2: HFTBench Daily Yield with std
> | Model parameter size |      Bitwidth Avg      |   Daily Yield(%)   |
> | :----------------------------: | :---------------------: | :---------------------: |
> | 14B(ours)                 |           7.2                |    26.59 ± 3.21           |
> | 14B                        |            8              |       22.14 ± 2.03        |
> | 14B                           |            16               |         18.01 ± 3.95        |
> | 7B                           |            16               |       -3.28  ±  1.37        |
> | 7B                           |            8               |     -6.93   ±   2.38         |
> | 7B                       |            8               |      -10.02   ±  3.81          |
>
> ## Disclaimer:
> We appreciate the reviewer’s disclaimer and the thoughtful attention to prior work.
>
> ### 1. We would like to clarify that, to the best of our knowledge, our work is indeed the first to systematically formulate and investigate the latency–quality trade-off for LLM agents operating in real-time environments.
>
> This assessment is also supported by other reviewers: several have explicitly acknowledged the novelty of our task formulation and benchmark design, with one reviewer describing the dual-benchmark setup as “groundbreaking.”
> ### 2. While Algorithm 1 may appear straightforward in isolation, it plays a central role in enabling our broader contribution, a practical and adaptive framework for latency-sensitive LLM inference.
>
> Prior work on model compression and mixed-precision typically focuses on **static throughput optimization or near-lossless accuracy**. In contrast, our FPX framework, summarized in Algorithm 1, introduces a **continuous, layer-wise mixed-precision assignment strategy** that is explicitly guided by **task-specific latency–reward trade-offs (see Section 4.1–4.2)**.
>
> What distinguishes our work is that latency is not just a systems constraint but is explicitly embedded in the reward function, as formalized in Equation 5 in our paper. Our benchmarks (HFTBench and StreetFighter) provide the **first concrete settings where latency directly impacts agent reward**, enabling us to evaluate FPX under realistic conditions.
>
> We believe this integration of calibrated precision control with reward-sensitive deployment constitutes both a novel problem formulation and a practical contribution that opens new directions in LLM agent optimization.
>
> ### 3. We appreciate the reviewer’s interest in the realism of our HFTBench design.
>  We would like to clarify that our high-frequency trading benchmark is **not a synthetic simulation, but rather a realistic backtesting environment** constructed from real-world per-second historical market data collected via Polygon.io (see Section 3.2, Lines 173–180).
> Agents in HFTBench receive synchronized 1-second tick data, including price levels and bid–ask spreads, and must decide whether and when to act. Crucially, execution prices are assigned using a linearly decaying function of response latency, which mimics the real-world queue-based price execution model common in modern exchanges. This design allows us to faithfully capture the interaction between latency and reward—where faster agents consistently gain more favorable prices.
>
> We believe this makes HFTBench a practically meaningful and realistic evaluation environment for latency-sensitive financial agents. It also opens the door for future extensions incorporating real multi-agent interaction or more granular tick-level data.

---

> > ### Comment · Reviewer_oRvQ · 2025-08-05
> >
> > I would like to thank the authors for addressing my concerns. I'm inclined to maintain my current (positive) rating for now, although it might be updated if necessary during the AC-reviewer discussion period.

---

> > > ### Author Response · Authors · 2025-08-06
> > >
> > > Dear Reviewer oRvQ,
> > >
> > > Thank you very much for your thoughtful feedback and for confirming that we have addressed your concerns. We also sincerely appreciate your current positive rating!
> > >
> > > Should you have any further questions or require additional clarification, we remain fully available during the discussion phase.
> > >
> > > To briefly reiterate, our paper makes the following contributions:
> > >
> > > 1. We present the first systematic analysis of the latency–accuracy trade-off for real-time LLM agents operating in latency-sensitive environments such as trading and gaming.
> > >
> > > 2. We introduce two new benchmarks, HFTBench and StreetFighter, built upon real market data and a real-time game engine, which enable more realistic evaluation of agent behavior under time pressure.
> > >
> > > 3. We propose FPX, an adaptive mixed-precision inference framework that continuously balances latency and accuracy based on both model characteristics and task-specific constraints.
> > >
> > > We believe these contributions offer valuable insights toward more deployable and efficient LLM agents, and can help guide future research in latency-critical decision-making.
> > >
> > > If you find these additions and clarifications compelling, we would be truly grateful if you would consider reflecting them in your final assessment.
> > >
> > > Best regards,
> > > Authors of "Win Fast or Lose Slow"

---

> ### Author Response · Authors · 2025-08-04
> **[Win fast lose slow] Thank You and a Quick Follow-Up**
>
> Dear Reviewer oRvQ,
>
> As the discussion phase is already well underway, we wanted to kindly check if there are any remaining concerns or questions we could help clarify.
>
> We’ve received some updated scores from reviewers after addressing their comments, and if you also feel that your concerns have been resolved, we would be sincerely grateful if you’d consider re-evaluating your score as well.
>
> Thank you again for your time and thoughtful feedback!
>
> Best regards, Authors of Win fast lose slow.

---

### Official Review · Reviewer_sJNR · 2025-07-03

**Clarity:** 3
**Significance:** 3
**Originality:** 2
**Rating:** 4
**Confidence:** 5

**Summary:**

This paper presents a timely and valuable investigation into the latency-accuracy trade-off for LLM-based agents in dynamic environments. The work addresses a critical gap in deploying LLMs for latency-sensitive applications like high-frequency trading and competitive gaming. The novel benchmarks (HFTBench and StreetFighter) and adaptive framework constitute a significant contribution to the field.

**Questions:**

Given the importance of calibration data choice, how sensitive is FPX to domain shift (e.g., calibrating on financial/gaming data)? A simple ablation study demonstrating this sensitivity would provide valuable practical insights and significantly benefit the community.

**Ethical Concerns:**

["NO or VERY MINOR ethics concerns only"]

**Final Justification:**

The authors provided additional experiments that directly address my two main concerns.

1. Robustness to calibration-data shift:
   • They re-calibrated FPX on the *Torchlight2* corpus (very different from Wikitext-2) and reported high Spearman correlations of the layer-sensitivity rankings (0.92 for Qwen-14B, 0.86 for Gemma-12B).
   • Downstream performance differences were small (≤ 2 pp win-rate on StreetFighter; ≤ 3 pp daily-yield on HFTBench), indicating that FPX is largely insensitive to domain shift in the calibration data.

2. Generalisability beyond Qwen models:
   • New results on the Gemma-3 family (4 B & 12 B) show consistent ELO improvements over FP8/FP16 baselines, confirming that FPX transfers to a distinct architecture.

These additions satisfactorily resolve my reservations about generality. I therefore keep my overall recommendation as *borderline-accept* (score = 4) but with strengthened confidence that the work will be useful to the community.

**Limitations:**

Yes

**Quality:**

3

**Strengths And Weaknesses:**

Strength
1.	Problem Significance: The paper convincingly establishes the importance of latency-sensitive decision-making for LLM agents (§1, §3), highlighting underexplored domains (trading/gaming) where delays directly impact rewards. This addresses a critical real-world deployment challenge.
2.	Both the proposed benchmark, framework and conclusion contribute to the community.



Weakness
Although the authors state that their evaluation is limited to the Qwen2.5 modes (ranging from 1.5B to 14B parameters) to ensure fair comparison and reduce search-space complexity, this raises significant concerns about the framework’s generalizability.

---

> ### Author Rebuttal · Authors · 2025-07-30
>
> ## Question1: How sensitive is FPX to domain shift?
>
> We thank the reviewer for the insightful question regarding the robustness of FPX calibration across data domains.
> To evaluate this, we conducted an ablation study comparing the layer-wise compression sensitivity rankings εₗ (defined in Algorithm 1 in our paper) generated by FPX under two distinct calibration datasets: Wikitext-2 (used in our main paper) and Torchlight2, a corpus of in-game dialogues and narrative text extracted from the action RPG [Torchlight II](https://github.com/hmi-utwente/video-game-text-corpora/tree/master/Torchlight%20II). Torchlight2 offers a very different text distribution compared to Wikitext, reflecting domain-specific content from real-time game environments.
>
> For both Qwen2.5-14B and Gemma3-27B, we observed **high Spearman rank correlations** between the layer rankings derived from the two datasets on the same model, **0.92 and 0.86**, respectively. This suggests that FPX yields stable and **transferable layer-wise precision assignments across domains**.
>
> Table R3T1 summarizes this result. In addition to correlation scores, we also compare downstream performance using the FPX strategy calibrated on Wikitext versus Torchlight2. We observe only **minor differences** in downstream task outcomes (e.g., win rate and daily yield), further reinforcing the **robustness of our approach to calibration domain shifts**. Win rate here denotes the outcome of a 40-round head-to-head match between two compressed models with different calibrated dataset. Each was calibrated using a different dataset (Wikitext vs. Torchlight) on the StreetFighter benchmark.
>
> ### Table R3T1: Ablation study of calibration datasets.
> | Model |      correlation      | Win rate| Daily Yield(wikitext) |Daily Yield(TorchLight)|
> | :------: | :---------------------: | :----------: | :------------------------: |:---------: |
> | Qwen-14B |       0.92       |    50%     |    26.52    | 24.17|
> | Gemma3-12B |   0.86   |     52.5%     |    31.30    | 33.05|
>
> We believe this empirical robustness is theoretically grounded:
> 1. The quantization error in low-bit inference is primarily governed by the range and distribution of activations and weights in each layer.
> 2. While input content may vary across datasets, the distributional characteristics of internal activations remain consistent per layer, especially in large pretrained LLMs.
> 3. Furthermore, the gap between FP8 and FP4 is large, and dominates small shifts in activation variation across domains. Thus, the ranking of compression tolerance is preserved.
>
> In contrast, sparsity-based compression may be more sensitive to input structure or task semantics. However, we intentionally avoid sparsity-based methods in this work due to their limited practical speedup, especially under realistic kernel implementations. In many latency-sensitive tasks, even perfect dynamic sparsity selection may not outperform well-optimized dense FP8 kernels.
>
> We view extending FPX to other compression paradigms and better calibration algorithms (e.g., structured sparsity) as an important avenue for future work.
>
> ## Weakness: Evaluation limited to the Qwen2.5 Models
> We thank the reviewer for raising this important concern regarding generalizability. To address this, we provide ELO ranking results for the **Gemma3 model family on the StreetFighter benchmark**. The evaluation setup follows exactly the same procedure described in Section 5 of the paper:
>
> We calibrate each model using the Wikitext dataset, apply FPX with compression ratio γ varying from 0.0 to 1.0 in steps of 0.1, and identify the best-performing γ for each model. Each model is then evaluated by playing 40 rounds of matches in the competitive tournament setting.
>
> These results demonstrate that FPX remains effective on non-Qwen models, improving both 4B and 12B variants of Gemma3 in terms of downstream win-rate (as reflected by ELO score). **This supports the generalizability of our framework across different architectures**.
>
> ### Table R3T2: Ablation study of Gemma3 family on Street Fighter.
> | Model parameter size |      Bitwidth Avg      | ELO score|
> | :----------------------------: | :---------------------: | :----------: |
> |4B(ours)                        |          6.8                 |    4.31       |
> |4B                                |             8                   |    2.19       |
> |12B(ours)                               |             16                   |    0.14       |
> |  4B                     |             6.4                   |    -1.32       |
> |12B                       |             8                   |    -2.14       |
> |12B                       |             16                   |    -2.97       |

---

> > ### Comment · Reviewer_sJNR · 2025-08-05
> >
> > The authors provided additional experiments that directly address my two main concerns.
> >
> > 1. Robustness to calibration-data shift:
> >    • They re-calibrated FPX on the *Torchlight2* corpus (very different from Wikitext-2) and reported high Spearman correlations of the layer-sensitivity rankings (0.92 for Qwen-14B, 0.86 for Gemma-12B).
> >    • Downstream performance differences were small (≤ 2 pp win-rate on StreetFighter; ≤ 3 pp daily-yield on HFTBench), indicating that FPX is largely insensitive to domain shift in the calibration data.
> >
> > 2. Generalisability beyond Qwen models:
> >    • New results on the Gemma-3 family (4 B & 12 B) show consistent ELO improvements over FP8/FP16 baselines, confirming that FPX transfers to a distinct architecture.
> >
> > These additions satisfactorily resolve my reservations about generality. I therefore keep my overall recommendation as *borderline-accept* (score = 4) but with strengthened confidence that the work will be useful to the community.

---

> ### Author Response · Authors · 2025-08-04
> **[Win fast lose slow]Thank You and a Quick Follow-Up**
>
> Dear Reviewer sJNR,
>
> As the discussion phase is already well underway, we wanted to kindly check if there are any remaining concerns or questions we could help clarify.
>
> We’ve received some updated scores from reviewers after addressing their comments, and if you also feel that your concerns have been resolved, we would be sincerely grateful if you’d consider re-evaluating your score as well.
>
> Thank you again for your time and thoughtful feedback!
>
> Best regards, Authors of Win fast lose slow.

---

### Official Review · Reviewer_2oJ3 · 2025-07-03

**Clarity:** 3
**Significance:** 3
**Originality:** 3
**Rating:** 4
**Confidence:** 3

**Summary:**

The authors address the latency–quality trade-off when deploying LLM agents in real-time environments. They (i) formalise latency-sensitive agent decision tasks, (ii) introduce two benchmarks—HFTBench for high-frequency trading and a StreetFighter gaming setup—and (iii) propose FPX, an adaptive mixed-precision framework that mixes FP8/FP4 at the layer level. FPX delivers smoother latency–accuracy Pareto curves than fixed-precision baselines, improving daily trading yield by ≈26 % and game win-rate by ≈80 % on their tasks .

**Questions:**

- While the paper provides a problem formulation, it lacks a unified evaluation metric. Metrics such as yield and latency or ELO and latency, are reported in separate columns. Do you consider introducing a normalized metric like reward ∕ latency to capture the trade-off?

- The proposed FPX relies on manual grid search over the compression ratio γ in discrete 0.1 steps. Can the agent be extended to support self-adaptation of γ at runtime?

**Ethical Concerns:**

["NO or VERY MINOR ethics concerns only"]

**Limitations:**

Yes

**Quality:**

2

**Strengths And Weaknesses:**

Strengths

- This work is the first to address latency-aware evaluation for LLM agents, offering a principled formulation of the problem

- The proposed FPX method outperforms fixed-precision baselines (FP8/FP16) across both tasks in terms of latency-normalized reward metrics, demonstrating a favorable speed–quality trade-off.

Weaknesses

- The StreetFighter evaluation is based on only 40 games per model pair, which may be insufficient for reliable ELO estimation
- Although the checklist (Q7) states that error bars are included, Table 1 and Table 2 present only point estimates, with no variance or confidence intervals reported

---

> ### Author Rebuttal · Authors · 2025-07-30
>
> ## Question1: Do we consider introducing a normalized metric like reward ∕ latency to capture the trade-off?
> We thank the reviewer for the insightful suggestion.
>
> We agree that having a unified evaluation metric is appealing in theory. However, in our setting, **the task-specific reward already implicitly incorporates latency**, making a normalized “reward/latency” ratio redundant and potentially misleading.
>
> Specifically:
> 1. In HFTBench, the **daily yield is directly impacted by latency through the price decay function** (Equation 5 in our paper), where slower decisions result in worse execution prices and thus lower profit.
> 2. In StreetFighter, the **win rate (and resulting ELO) reflects the agent's ability to respond quickly and accurately in a real-time environment**. Again, latency is implicitly encoded in the game outcome.
>
> Therefore, both reported metrics, daily yield and ELO, are already **final metrics that reflect the integrated effect of latency and accuracy for their respective tasks**.
>
> Moreover, we believe that a single normalized metric **may not generalize well across tasks**, since different latency-sensitive applications have **different latency-reward dynamics** (discussed in Section 5.3 and 5.4). A naive reward/latency ratio may fail to capture these nuances.
>
> We agree that defining a general-purpose trade-off metric is an interesting direction for future work, especially for comparing across domains.
>
> ## Question2: Can the agent be extended to support self-adaptation of γ at runtime?
>
> We thank the reviewer for this valuable question. We confirm that **FPX can be extended to support runtime adaptation of the compression ratio γ**.
>
> At a system level, this is feasible by storing multiple precision variants of the model or key layers, and dynamically selecting between them during inference. This allows the agent to change γ on the fly depending on the environmental context.
> To demonstrate this, we implemented a simple rule-based policy in the StreetFighter benchmark:
>
> 1. When the two players are far apart, precision is less critical, so the agent uses the faster FP8 model.
> 2. When the players are close (within two action steps), the agent switches to the FPX-compressed model (γ = 0.3), which offers higher accuracy for fine-grained combat decisions.
>
> We evaluated this self-adaptive model against fixed-γ models using 40-round tournaments, and report the number of wins in Table R2T1.
>
> ### Table R2T1: Ablation study of self-adaptive and fixed trade-offs.
> | Model parameter size |      Self-adpt Win Round      |  Fixed Win Round      |
> | :----------------------------: | :---------------------: | :---------------------: |
> | 3B                           |           25                |           15              |
> | 7B                           |            22              |           18              |
> | 14B                           |            20               |           20              |
>
> We found out that **self-adaptive is beneficial to most of the models**. But when the latency of response is far from latency action ceiling(200ms), the benefit of adaptive strategy is eliminated.
>
> ## Weakness1: The StreetFighter evaluation is based on only 40 games per model pair, which may be insufficient for reliable ELO estimation.
> We appreciate the reviewer’s concern regarding the statistical reliability of our ELO estimation.
>  To address this, we conducted an extended evaluation using 400 rounds of head-to-head matches per model pair on the StreetFighter benchmark.
>
> The updated results are shown below in Table R2T2. We observe that:
>
> 1. The relative ranking of models remains consistent with our original 40-round results.
> 2. The variance across blocks of 40 games is low, indicating that our earlier estimates were already directionally stable.
>
> ### Table R2T2: ELO Scores Based on 400-Game Tournaments (StreetFighter)
> | Model parameter size |      Bitwidth Avg      |  ELO(400 rounds)   |  ELO(40 rounds)   |
> | :----------------------------: | :---------------------: | :---------------------: | :---------------------: |
> | 3B(ours)                 |           6.8                |           5.91              |    5.93                      |
> | 7B（ours)               |            7.2              |           2.31              |      2.33                    |
> | 3B                           |            8               |           2.23              |      2.19                         |
> | 3B                           |            16               |           0.27             |      0.25                         |
> | 7B                           |            8               |           -0.44             |      -0.44                        |
> | 1.5B                       |            8               |           -1.27             |      -1.25                        |
>
> ## Weakness2: Error bar missing.
> We thank the reviewer for pointing this out. You are correct. Our initial submission reported single-trial results without error bars, and we acknowledge this mistake in answering Checklist Question 7.
>
> To address this, we have now repeated StreetFighter experiments 10 times(400 rounds in total), HFTBench 3 times, and report mean ± standard deviation across these runs. We will incorporate these revised results (with appropriate error bars) into the final version of the paper.
>
> ### Table R2T3: StreetFighter ELO Scores with std error
> | Model parameter size |      Bitwidth Avg      |   ELO(40 rounds)   |
> | :----------------------------: | :---------------------: | :---------------------: |
> | 3B(ours)                 |           6.8                |    5.92 ± 0.13            |
> | 7B（ours)               |            7.2              |       2.31 ±  0.17         |
> | 3B                           |            8               |         2.21  ±  0.05        |
> | 3B                           |            16               |       0.26  ±  0.09        |
> | 7B                           |            8               |     -0.44   ±   0.12         |
> | 1.5B                       |            8               |      -1.26   ±  0.07          |
> ### Table R2T4: HFTBench Daily Yield with std error
> | Model parameter size |      Bitwidth Avg      |   Daily Yield(%)   |
> | :----------------------------: | :---------------------: | :---------------------: |
> | 14B(ours)                 |           7.2                |    26.59 ± 3.21           |
> | 14B                        |            8              |       22.14 ± 2.03        |
> | 14B                           |            16               |         18.01 ± 3.95        |
> | 7B                           |            16               |       -3.28  ±  1.37        |
> | 7B                           |            8               |     -6.93   ±   2.38         |
> | 7B                       |            8               |      -10.02   ±  3.81          |

---

> ### Author Response · Authors · 2025-08-04
> **[Win fast lose slow]Thank You and a Quick Follow-Up**
>
> Dear Reviewer 2oJ3,
>
> As the discussion phase is already well underway, we wanted to kindly check if there are any remaining concerns or questions we could help clarify.
>
> We’ve received some updated scores from reviewers after addressing their comments, and if you also feel that your concerns have been resolved, we would be sincerely grateful if you’d consider re-evaluating your score as well.
>
> Thank you again for your time and thoughtful feedback!
>
> Best regards, Authors of Win fast lose slow.

---

> ### Author Response · Authors · 2025-08-06
>
> Dear Reviewer  2oJ3,
>
> Thank you again for your effort reviewing our paper and for your valuable feedback. Since the discussion period is ending soon, we would like to follow up to see if our rebuttal has addressed your concerns. We would be happy to provide any further clarification needed to earn your reconsideration of our work. Thank you very much.
>
> Sincerely,
>
> Authors of Submission 17147

---

### Official Review · Reviewer_AkfL · 2025-07-18

**Clarity:** 4
**Significance:** 2
**Originality:** 3
**Rating:** 4
**Confidence:** 2

**Summary:**

This paper establishes the first formal framework for latency-quality trade-offs in LLM agents operating in dynamic environments. It introduces "latency-sensitive agent decision tasks" where delayed actions incur real-time penalties , and contributes: (1) Two novel benchmarks—HFTBench (high-frequency trading simulator with Polygon.io data) and StreetFighter (real-time gaming platform); (2) FPX, an adaptive mixed-precision framework using layer-wise FP4/FP8 inference via offline calibration;

**Questions:**

- Threshold sensitivity: StreetFighter’s 200ms action ceiling appears game-specific. How would results scale to 100ms (e.g., autonomous driving) or 500ms (e.g., robotic control)?

- MoE compatibility: Does FPX maintain efficacy for mixture-of-experts architectures?

- The current experiment only tests a single agent, but the real trade/game involves a multi-party game. Is there a "delayed arms race" when multiple FPX agents are in parallel?

**Ethical Concerns:**

["NO or VERY MINOR ethics concerns only"]

**Limitations:**

Authors acknowledge layer-wise control constraints. Additional discussion needed on: Multi-agent latency competition dynamics.

**Quality:**

3

**Strengths And Weaknesses:**

Experimental rigor is outstanding: granular ablation across Qwen2.5 models, Addresses a critical gap where prior agent frameworks ignore latency penalties. Benchmarks enable standardized evaluation for real-time AI systems. The task formulation and dual-benchmark design are groundbreaking.

---

> ### Author Rebuttal · Authors · 2025-07-30
>
> ## Question1: Threshold sensitivity study
> We thank the reviewer for the thoughtful question regarding the action latency ceiling and its task dependence.
> Task-specific ceilings are realistic and often reflect physical or system-level constraints.
>
>  In real-time environments like StreetFighter, planning latency beyond the 200ms window has diminishing returns, since **actions cannot be executed faster than the environment permits**. This aligns with the principle that planning should not outpace action, which holds in many closed-loop decision systems such as robotics and gaming.
>
>
> In contrast, tasks like high-frequency trading behave differently: decisions can be executed nearly instantaneously, and latency directly determines priority in the execution queue. In such ranking-dominant tasks, **lower latency is always beneficial**, even far below any implicit ceiling. Our framework accommodates both settings by allowing compression to push latency toward the most beneficial region for the task.
>
> While we were unable to build new datasets with stricter or looser latency ceilings due to time constraints, we can indirectly study this question via **hardware variation**. The effective inference latency of each model varies by hardware, and thus changes how close each model comes to the latency ceiling.
>
>
> To explore this, we evaluate the StreetFighter benchmark on NVIDIA H200, measuring ELO scores of FPX-compressed models across different Qwen3 model sizes. All models are evaluated using the same game rules (200ms ceiling), but their inference latency on H200 differs significantly. The results are shown in Figure R1T1, demonstrating how model hardware combinations influence outcomes relative to the ceiling. The reasoning that Qwen3-14B fp8 is the top winner is that the **latency is near 200ms at H200**. Though 4B and 8B models are faster than 14B models, they are not the top winners due to the **gaming latency ceiling**.
> ### R1T1 StreetFighter with Qwen2.5 models on H200
> | Rank | Model Size | Bitwidth | Score |
> |------|------------|----------|-------------------|
> | 1    | 14B_8bit   | 8        | **0.824**         |
> | 2    | 32       | 16       | **0.549**         |
> | 3    | 14B        | 16       | **0.315**         |
> | 4    | 4B_8bit    | 8        | **0.299**         |
> | 5    | 32B        | 16       | **0.114**         |
> | 6    | 8B         | 16       | **0.025**         |
>
> ## Question2: Does FPX maintain efficacy for mixture-of-experts architectures?
> We thank the reviewer for the interesting question regarding FPX compatibility with mixture-of-experts (MoE) architectures.
>
>  To evaluate this, we conducted an initial experiment using Qwen3-30B-A3B, a MoE model with 128 experts (2 active per forward pass). We applied our FPX calibration procedure as usual, assigning different precision levels to transformer blocks and expert layers independently. The experiment was run on 3 RTX 5090 GPUs, and follows the same evaluation setup used for the standard FPX StreetFighter benchmark. We found that prefilling time for such large moe models on 5090 is too time-consuming. So we apply prefill caching with FPX as well. Also, to embed FPX with MOE models, we see all experts in FFN layers as one united operator. We rank and compress them together at the same time.
>
> The results are shown below in Table R1T1, and demonstrate that FPX continues to provide strong latency–quality trade-offs even in MoE models.
>
> ### Table R1T2: MOE model on StreetFighter
> | Model |      Bitwidth Avg      |   Win Rate vs. FP16  |
> | :----------------------------: | :---------------------: | :---------------------: |
> | Qwen3-30B-A3B(ours)                 |           6.4                |    92.5%            |
> | Qwen3-30B-A3B                 |           8                |    82.5%            |
> | Qwen3-30B-A3B                 |           16                |    50%            |
>
> These results demonstrate that FPX remains effective in the MoE setting, providing a substantial improvement in win rate (92.5%) compared to both FP16 and FP8 baselines, while achieving lower average bitwidth.
>  This suggests that FPX can exploit the sparsity and modularity of MoE architectures to apply precision selectively, further enhancing both latency and decision quality.
> We consider extending FPX with expert-aware calibration and dynamic routing–precision interaction an exciting direction for future work.
>
> ## Question 3 and limitation: multiple FPX agents competition
> We appreciate the reviewer’s question and would like to clarify that our current evaluation **already models a multi-agent competitive setting**, but has not been tested only with a single agent.
>
>  In the StreetFighter benchmark, there are **two agents** competing with each other. Each agent controls a player in the game and runs inference **independently** on its own hardware and model.
>
> This naturally reflects a “delayed arms race” dynamic: faster agents react more quickly and gain the upper hand, while slower ones are penalized. In fact, the incentive to reduce latency to outcompete an opponent is a key driving force behind our formulation.
> That said, we acknowledge that fully simulating multiple FPX agents running in parallel on heterogeneous hardware, with timing feedback loops and cross-agent adaptation, would further enrich this evaluation. Due to current hardware constraints, **we leave multi-agent FPX interaction under joint optimization over two players as an exciting direction for future work**.
>
> We believe that when multiple agents are deployed collaboratively or competitively, the latency–accuracy trade-off landscape becomes significantly more complex. In such settings, each agent’s optimal behavior may depend not only on its own constraints but also on the timing and strategy of others. This calls for the development of more sophisticated, coordination-aware algorithms that can adaptively control each agent’s compression level at runtime based on external observations or shared signals.

---

> ### Author Response · Authors · 2025-08-04
> **[Win fast lose slow] Thank You and a Quick Follow-Up**
>
> Dear Reviewer AkfL,
>
> As the discussion phase is already well underway, we wanted to kindly check if there are any remaining concerns or questions we could help clarify.
>
> We’ve received some updated scores from reviewers after addressing their comments, and if you also feel that your concerns have been resolved, we would be sincerely grateful if you’d consider re-evaluating your score as well.
>
> Thank you again for your time and thoughtful feedback!
>
> Best regards,
> Authors of Win fast lose slow.

---

> ### Author Response · Authors · 2025-08-06
>
> Dear Reviewer  AkfL,
>
> Thank you again for your effort reviewing our paper and for your valuable feedback. Since the discussion period is ending soon, we would like to follow up to see if our rebuttal has addressed your concerns. We would be happy to provide any further clarification needed to earn your reconsideration of our work. Thank you very much.
>
> Sincerely,
>
> Authors of Submission 17147

---

### Author Response · Authors · 2025-08-06
**[Win fast or lose slow]Friendly Reminder, Discussion Phase Nearing Completion.**

Dear Reviewers,

As the discussion phase draws to a close, we would like to sincerely thank you, especially for those who have already engaged in the discussion and expressed that our response addressed your concerns. We truly appreciate the thoughtful feedback and the updates to your scores and confidence.

If there are any further questions or remaining concerns, we are fully available and happy to provide clarifications.

For reviewers who have not yet participated in the discussion, we would greatly appreciate your input and look forward to your comments. Your perspective would be extremely valuable in helping us further improve the work.

We are deeply grateful for all of your suggestions and for your support of the academic community.

To briefly reiterate, our submission is, to the best of our knowledge, the first to:
1. **Systematically evaluate the latency–accuracy trade-off in real-time LLM agent** decision-making under latency-sensitive environments such as trading and gaming.
2. **Introduce two realistic latency-sensitive benchmarks**, built on real market data and interactive game engines.
3. **Propose FPX**, a mixed-precision inference framework that **continuously trades off latency and accuracy**, tailored to both model structure and task-specific constraints.

We believe these contributions can help guide future research on the deployment of LLM agents in latency-critical settings, and offer insights for both the academic and applied ML communities.

Thank you again for your time and effort in reviewing our paper.

Best regards,
Authors of "Win Fast or Lose Slow"

---

### Note · Authors · 2025-08-13

We thank the AC and reviewers for their time and effort in overseeing the review process and discussion. We provide a final remark at the end of the rebuttal period, including a summarization of our paper and rebuttal.
### **Summarization of our Contribution**
To our best knowledge, this is the **first work** to systematically formulate and study the **latency–accuracy trade-off for LLM-based agents** in **real-time, latency-sensitive decision-making tasks**. While prior agent and LLM research mainly targets **static domains** such as coding or search, where latency has **negligible effect** on reward, many real-world applications, from high-frequency trading to gaming, have **intrinsic latency constraints** where delayed actions directly degrade performance.

To address this gap, we define the new class of **latency-sensitive agent decision tasks**, where **reward depends jointly on decision quality and timeliness**. We introduce two realistic benchmarks, HFTBench and StreetFighter.

To study the trade-off of latency and accuracy, we propose FPX, an adaptive mixed-precision inference framework that dynamically combines FP8 and FP4 at the layer level based on quantization tolerance, **enabling continuous, fine-grained control of latency–accuracy trade-offs without retraining**. FPX includes a specially designed kernel that removes conversion overhead, ensuring gains in precision on real hardware.

Our results show that FPX achieves substantial downstream improvements, demonstrating that controlled accuracy loss can be beneficial when latency is part of the reward. We believe this work establishes a **foundation for future research on latency-aware evaluation**, algorithm design, and deployment of LLM agents, **bridging the gap between theoretical model efficiency and real-world decision performance**.

### **Summarization of Rebuttal**
During the rebuttal, we addressed all reviewer concerns with additional experiments, clarifications, and new results. We confirmed that all FP4/FP8 results were obtained on real hardware (RTX 5090) and added GB200 evaluations, showed FPX’s effectiveness on MOE and Gemma models, clarified novelty beyond standard mixed precision, and demonstrated dynamic γ tuning integrated with task dynamics.

Most reviewers acknowledged that their concerns were resolved, with several explicitly increasing their scores or confidence. We believe the paper now presents a clear, well-validated, and impactful contribution to the community.

---

### Decision · Program_Chairs · 2025-09-17

**Decision:**

Accept (spotlight)

**Comment:**

This work contributes the first study on the tradeoff between latency and response generation for downstream task completion, together with the introduction of a couple benchmark environments for future investigation. The reviewers unanimously appreciated the framework and contributed insights, which justifies acceptance at this time.